# Plant-Derived Substances in the Fight Against Infections Caused by *Candida* Species

**DOI:** 10.3390/ijms21176131

**Published:** 2020-08-25

**Authors:** Ibeth Guevara-Lora, Grazyna Bras, Justyna Karkowska-Kuleta, Miriam González-González, Kinga Ceballos, Wiktoria Sidlo, Maria Rapala-Kozik

**Affiliations:** 1Department of Analytical Biochemistry, Faculty of Biochemistry, Biophysics and Biotechnology, Jagiellonian University in Krakow, Gronostajowa 7, 30–387 Krakow, Poland; ibeth.guevara-lora@uj.edu.pl (I.G.-L.); kinga.ceballos@student.uj.edu.pl (K.C.); 2Department of Comparative Biochemistry and Bioanalytics, Faculty of Biochemistry, Biophysics and Biotechnology, Jagiellonian University in Krakow, Gronostajowa 7, 30–387 Krakow, Poland; grazyna.cholewa@uj.edu.pl (G.B.); justyna.karkowska@uj.edu.pl (J.K.-K.); miriam.gonzalez.gonzalez@doctoral.uj.edu.pl (M.G.-G.); wiktoria.sidlo@student.uj.edu.pl (W.S.); 3Institute of Zoology and Biomedical Research, Faculty of Biology, Jagiellonian University in Krakow, Gronostajowa 9, 30–387 Krakow, Poland

**Keywords:** *Candida* spp., antifungal treatment, plant extracts, plant metabolites

## Abstract

Yeast-like fungi from the *Candida* genus are predominantly harmless commensals that colonize human skin and mucosal surfaces, but under conditions of impaired host immune system change into dangerous pathogens. The pathogenicity of these fungi is typically accompanied by increased adhesion and formation of complex biofilms, making candidal infections challenging to treat. Although a variety of antifungal drugs have been developed that preferably attack the fungal cell wall and plasma membrane, these pathogens have acquired novel defense mechanisms that make them resistant to standard treatment. This causes an increase in the incidence of candidiasis and enforces the urgent need for an intensified search for new specifics that could be helpful, alone or synergistically with traditional drugs, for controlling *Candida* pathogenicity. Currently, numerous reports have indicated the effectiveness of plant metabolites as potent antifungal agents. These substances have been shown to inhibit growth and to alter the virulence of different *Candida* species in both the planktonic and hyphal form and during the biofilm formation. This review focuses on the most recent findings that provide evidence of decreasing candidal pathogenicity by different substances of plant origin, with a special emphasis on the mechanisms of their action. This is a particularly important issue in the light of the currently increasing frequency of emerging *Candida* strains and species resistant to standard antifungal treatment.

## 1. Introduction

Humans have co-evolved with millions of fungal species [1] of which only several hundred can be infectious, most severely affecting individuals with immature or defective immune system―neonates and elderly, individuals with hematological malignancies, acquired immunodeficiency syndrome or congenital immunodeficiencies, and patients after prolonged treatment with broad-spectrum antibiotics. Paradoxically, the development of advanced medical therapies could predispose some patients to fungal infections that affect individuals under cancer care, autoimmune disease treatments, trauma and intensive care, organ and hematopoietic stem cell transplantations, and patients that have been subjected to surgery and the use of parenteral nutrition, intravenous catheters or mechanical ventilation [2,3,4,5,6,7].

Commonly identified fungal pathogens in humans are *Candida* species―usually commensal yeasts that constitute a part of physiological microbiota, mildly colonizing human skin, oral cavity, vagina, and gut, but changing into the opportunistic pathogens under conditions of inefficient host immunity [8]. The worldwide distribution, plasticity in adapting to new challenging niches, and diversity of species and strains of *Candida* genus determine that these yeast-like fungi cause a wide range of different types of infections in susceptible individuals [9,10,11]. These might be infections with high morbidity but not life-threatening, such as painful and itchy mucocutaneous candidiases including candidal vulvovaginitis and balanitis, oropharyngeal candidiasis, internal overgrowth in the gastrointestinal tract, keratitis or skin and nail mycoses [12,13,14,15,16]. However, currently, a serious medical problem still remains with severe invasive fungal diseases associated with candidemia and deep-seated candidiases of internal organs, characterized by the frequent nosocomial origin and high mortality rates reaching 30–45% of infected patients [17,18,19,20,21,22,23]. Of more than two hundred species of the *Candida* genus, about twenty present important medical significance, with most prevalent *C. albicans* that accounts for nearly half of the diagnosed superficial and systemic candidiases in humans [24,25]. In addition to *C. albicans*, the other main etiological agents of candidal infections include *C. glabrata*, *C. parapsilosis*, *C. tropicalis*, *C. krusei*, *C. dubliniensis*, *C. kefyr*, *C. guilliermondii*, *C. lusitaniae* and the currently globally emerging *C. auris*, that all should be paid particular attention to because of their diverse virulence strategies and increased resistance to the antifungals used [20,26,27]. The distribution of particular species among different groups of susceptible patients varies noticeably depending on the geographical region and specific risk factors predisposing to candidal infection that include a variety of conditions based on the weakening of the host immune defense and shifting the balance from the harmless commensal fungus to the dangerous pathogen. The currently recorded steady increase in the incidence of infections caused by non-albican *Candida* species is in particular noteworthy [28,29]. Among this group of *Candida* fungi, the most common cause of invasive candidiasis in 2006–2016 was *C. glabrata* in North America, Asia-Pacific region and Europe, whereas it was *C. parapsilosis* in Latin America; and the second in frequency was *C. parapsilosis* in Europe and North America and *C. tropicalis* in Asia–Pacific region and Latin America [30].

## 2. Virulence Factors of *Candida* Species to be Considered for the Development of New Antifungal Strategies

The successful colonization and occasional invasion of the human host are achieved by *Candida* species through their diversified virulence and survival strategies, suited to the conditions in the host organism. Of the highest importance is a tremendous facility of *C. albicans* cells for morphological transition from yeast-like round cells to elongated pseudohyphae and filamentous hyphae [31]. Each of these forms is equipped with a set of suitable adhesins or other surface-located proteins, enabling the further penetration, dissemination, and persistence of pathogens in host tissues [32]. The main adhesins mediate the adherence of *C. albicans* cells to abiotic surfaces, other microorganisms, and host cells belonging to the agglutinin-like sequence (Als) protein family of glycosylphosphatidylinositol (GPI)-linked cell surface glycoproteins [33,34], encoded by eight *ALS* genes (*ALS1–7* and *ALS9*). Another important adhesin is Hwp1―a hypha-associated GPI-linked protein involved in the direct attachment of fungal cells to host epithelial cells and in the formation of drug-resistant biofilms [35]. Moreover, other GPI-linked proteins such as Eap1, Iff4 and Ecm33 support *C. albicans* cells’ adhesion to host tissues. A similar effect was also observed for some proteins that are non-covalently associated with the cell wall such as putative β-glucanase Mp65 and β-1,3-glucanosyl transferase Phr1 as well as for the integrin-like surface protein Int1 [36].

The adhesion also triggers the possible formation of biofilms―highly complex microbial structures, firmly fixed within an extracellular matrix and tightly attached to various surfaces. The biofilm ensures resistance to antimicrobial agents and host immune factors, not only due to a tight architecture of the formed fungal cell community but also by the increased expression of drug efflux pumps and the formation of biofilm matrix, composed by glucans, mannans, proteins and nucleic acids, providing the protection and metabolic plasticity [37,38]. The major components of the biofilm matrix, β-1,3-glucans, are regulated by the zinc-responsive transcription factor Zap1 and glucoamylases (Gca1 and Gca2), glucan transferases (Bgl2 and Phr1) and exoglucanase Xog1 [39,40].

Another virulence trait is related to secreted extracellular hydrolases, e.g., phospholipases that degrade phospholipids of host cell membranes and facilitate tissue penetration by fungal cells. These enzymes are encoded by genes *PLB1-5,* and their contribution to *Candida* pathogenicity was documented e.g., by attenuation in virulence of mutant strains with selective elimination of a particular *PBL* gene [41]. Phospholipases are also associated with antifungal resistance and biofilm production [42]. Similar effects of reduced virulence of mutant strains were also noted for the deletion of genes *LIP1-10* encoding the members of the lipase family [43].

Furthermore, aspartyl protease isoenzymes (*SAP1-10*) can be used by fungus in the damage of epithelial and mucosal barrier proteins such as collagen, keratin, and mucin. Additionally, they also degrade immunoglobulins, complement, cytokines and antifungal peptides [44,45,46,47] and can activate the production of mediators of inflammation—kinins—bypassing the host control and facilitating the invader spreading in the host tissues [48].

## 3. Current Methods for Treatment and Management of Superficial and Invasive Candidiases

Despite the advances in the medical treatment of mycoses, specific therapies are limited by similarities in the structure and biochemical processes of human and fungal cells. Therefore, the most effective drugs are based on the differences in the composition of the plasma membrane occurring between host’s and pathogen’s cells, as well as on the cell wall formed exclusively by yeasts. Other antifungals block nucleic acids or protein synthesis. For the treatment of *Candida* infections, four most popular classes of drugs are currently used, namely echinocandins, azoles, polyenes and fluoropyrimidines (Figure 1) [49].

The first class of the most effective anticandidal drugs—echinocandins—contains semisynthetic cyclic lipopeptides derived from natural fungal echinocandins that alter the cell wall biosynthesis by inhibition of β-1,3-glucan synthase. Their action resulted in the reduction of the fungal cell wall strength [50], increased osmotic instability and increased susceptibility to cell lysis [51]. Echinocandins include caspofungin, micafungin and anidulafungin [52] and exhibit high activity against *C. albicans*, *C. tropicalis, C. glabrata* and *C. krusei.* However, *C. parapsilosis* is intrinsically less susceptible to these drugs due to a naturally occurring alteration in the gene encoding FSK1 subunit of β-1,3-glucan synthase [53]. Resistance of a similar origin was also identified for some *C. glabrata* and *C. krusei* strains [54] in which the mutations in *FSK1* and *FSK2* genes were observed [55]. For C. *auris* isolates it was demonstrated that the prevalence of resistance was 3.8%, i.e., of a similar order as that for *C. glabrata* [56]. Echinocandins are also effective in the prevention of fungal biofilm formation [57] and less toxic to human cells, thus being frequently used for first-line therapy in the treatment of invasive candidiasis [58].

The next group of antifungals comprises drugs causing the alteration of the membrane function such as polyenes represented by nystatin, natamycin and amphotericin B (AMB). All of them are natural components isolated from bacteria from the *Streptomyces* genus. AMB is produced by *S. nodosus* [59], natamycin by *S. natalensis, S. chattanoogensis, S. lydicus* and *S. gilvosporeus* [60], and nystatin by *S. noursei* [61]. These highly lipophilic macrolides are able to penetrate the phospholipid bilayer of the plasma membrane, in which they bind to ergosterol and promote the formation of transmembrane channels resulting in cellular ionic imbalance and leading to cell death [62]. Although all *Candida* species are susceptible to polyenes, the treatment of candidiasis caused by *C. krusei* and *C. glabrata* requires a maximal safe dose of drugs; however, the usage of polyenes is limited by their nephrotoxicity [58]. This adverse effect results from the affinity of polyenes to cholesterol located in the membrane of host cells, affecting the permeability of the renal tubules [63,64]. The best-known drug from this group—AMB—is also effective against *Candida* spp. biofilm [57] and is often used in a safer liposome-encapsulated form; however, this form is also more expensive, making this drug medication for second-line therapy [65]. Moreover, nystatin or natamycin are useful only for the treatment of superficial infections [60,66].

Although rare, the resistance to polyenes was described for some *Candida* strains isolated from infected patients, with fungal cells overcoming polyene action by reducing the amount of ergosterol in the plasma membrane. This is implemented by some defects in the *ERG3* and *ERG6* genes encoding enzymes of ergosterol biosynthesis, or its substitution by non-ergosterol cytoplasmic sterols and lipids [65,67].

Azoles belong to the fungistatic drugs targeting the cell membrane that inhibit an enzyme essential for ergosterol biosynthesis. These drugs act by binding and inhibition of the lanosterol-14α-demethylase, a cytochrome P450 enzyme (the product of *ERG11* gene) mediating a rate-limiting step in ergosterol biosynthesis. This results in the formation of toxic sterols (14-α-methylsterols), placed loosely within lipid bilayers and decreasing tightness and stability of fungal cell membrane [59]. This group of synthetic fungistatic drugs includes imidazoles (clotrimazole, econazole, miconazole and ketoconazole), used primarily for the treatment of superficial infections, and triazoles (fluconazole, itraconazole, voriconazole, posaconazole, isavuconazole) preferred for fighting against invasive candidiasis [65,66,68]. Although generally well-tolerated, azoles also have several limitations, such as hepatoxicity, mainly observed for clotrimazole and miconazole [69]. Moreover, there are problems with azole efficiency. They are not active against *Candida* spp. biofilm [57]; *C. krusei* possesses an intrinsic resistance to azoles and *C. glabrata* is less susceptible to them [66]. Additionally, the emerging resistance to this group of drugs among isolates of *Candida* spp., especially *C. auris*, *C. parapsilosis* and *C. tropicalis* is currently observed [54,70] as a consequence of the massive preventive application of fluconazole [71,72] and common use of agricultural fungicides, structurally similar to azoles [73]. The resistance relies on the mutations or overexpression of *ERG11* and *ERG3* genes, which lead to increased tolerance to methylated sterols and overexpression of drug-efflux pumps that transport azoles outside of the fungal cell [65,67].

The last group of antifungals active against *Candida* spp. contains the factors that block the synthesis of fungal nucleic acids and are represented by synthetic analogs of fluorinated pyrimidine. The most commonly used is 5-fluorocytosine that enters the fungal cell via the transport mechanism based on cytosine permease, and inside the fungal cell it is converted to 5-fluorouracil that, after phosphorylation, can inhibit thymidylate synthase, essential during DNA replication. After a further phosphorylation step, the compound formed is embedded in the RNA structure, causing premature chain termination [62,65]. The disadvantages of such treatment are the adverse side effects of using 5-fluorocytosine, including the hepatic impairment, interference with bone marrow function and rapid development of resistance by *Candida* species [68]. The resistance observed can be due to a deficiency of enzymes involved in 5-fluorocytosine transport and metabolism [65,67] or overproduction of natural substrate for thymidylate synthase [74]. Moreover, this type of drug is not effective against *C. krusei*, which possesses an intrinsic resistance [66]. As a consequence, 5-fluorocytosine is applied for patients at low concentrations and in combination with other antifungals [58], especially with AMB [68]. Currently, an important medical problem has been generated due to the development of multidrug resistance by individual *Candida* strains and species, as is the case of the emerging pathogen *C. auris* [75]. The *Candida* resistance to antifungals is associated with the modulation of drug targets during the fungal morphological switch, the increase in the abundance and activity of drug exporters, acquired resistance to environmental stress and biofilm formation. The harmful side effects of currently used drugs, the increase in resistance to applied treatments, and formation of the protective biofilm structure demonstrate an urgent need for directly targeting and controlling different fungal virulence factors and selecting new types of drugs to limit the resistance as an effective method of combating fungal infection.

## 4. A New Perspective for Plants’ Components in Antifungal Therapy

Recent reports indicate that plants are currently a promising source of substances with potent fungicidal effects. Taking into consideration currently increasing drug resistance in *Candida* spp., intensive studies have been performed to discover new substances which may be helpful in candidiasis prevention and in the eradication of fungal pathogens. Especially in the last five years, new achievements in this field have demonstrated an interest in the anti-*Candida* activity of plant-derived crude extracts or isolated phytochemicals. A unique feature of products extracted from plants is their high structural diversity, which includes primary and secondary metabolites. A wide range of compounds isolated from medicinal plants has been tested for the inhibition of *Candida* cell growth, adhesion, hypha formation, and restriction of biofilm formation or destruction of mature biofilms. Primary anticandidal metabolites are some peptides and lipids, while secondary metabolites with anticandidal activities include alkaloids, terpenes, steroids, phenolic compounds (flavonoids, tannins and phenolic acids) and other organic substances [76,77,78,79,80,81,82].

Although a great number of reports have demonstrated the antifungal effect of plant extracts during candidiasis treatment, the mechanisms associated with their anticandidal activity are still not satisfactorily understood. In this work, we summarize the current research focused not only on the discovery of new plant components with antifungal potential but also on the mechanisms responsible for their effect on the inhibition of growth and virulence of *Candida* spp. Therefore, two distinct approaches are herein described. One concerns plant material-related mechanisms that affect the cell wall, plasma membrane, and biofilm extracellular matrix of the most commonly isolated *Candida* species, and the second, the action of plant extracts in the regulation of cellular functions, including those related to fungal virulence.

### 4.1. Mechanisms of the Action of Plant Metabolites on the Fungal Cell Wall, Plasma Membrane and Biofilm Extracellular Matrix

The cell wall and plasma membrane of *Candida* cells are the first targets for substances with antifungal potential. Moreover, the extracellular matrix of fungal biofilms contributes significantly to candidal virulence thus becoming a promising object for anticandidal treatment.

As the drug resistance in *C. albicans* is essentially multifactorial, and the production of biofilm matrix is one of the relevant resistance-related mechanisms associated with the development of biofilm, pharmacological interventions during the early phase of biofilm formation leading to the reduction of its extracellular matrix are strongly correlated with increased treatment effectiveness [83]. Numerous reports have indicated that the mechanisms responsible for the reduction in carbohydrate content of biofilm extracellular matrix are associated with changed expression of gene encoding enzymes modifying the amount of β-1,3-glucan (*FKS1, BGL2P, PHR1P, XOG1)*, β-1,6-glucan (*BIG1, KRE5*) and mannan (*PMR1*, *ALG11*, *MNN9*, *MNN4-4*, *VRG4*, *VAN-1)* [83,84].

In the last few years, several studies related to extracellular matrix destabilization by plant-derived substances have been performed. Samel reports have shown reduced amounts of polysaccharides in the extracellular matrix of *Candida* spp. biofilm after treatment with essential oils from *Cinnamomum* [85], and ethanolic extracts from *Aucklandia lappa* [86] and *Paeonia lactiflora* [87]. Recently, a decreased production of polysaccharides and extracellular DNA by fungal biofilm under the influence of plant-derived substances—the alkaloid chelerythrine from Ranunculales and the terpene sanguinarine from *Zingiber zerumbet*—was described [88]. Although these studies demonstrated a significant decrease in the carbohydrate content within the fungal cell wall as a result of the action of plant-derived substances, no specific mechanisms have been elucidated so far. An interesting report with terpene cinnamaldehyde showed its fungicidal activity against *C. glabrata* isolates, demonstrating reduced expression of the *FKS1* gene responsible for β-1,3-glucan biosynthesis. However, in that study, the content of extracellular carbohydrates was actually observed to increase [89]. On the other hand, a different course of action for β-1,3-glucan reduction by poacic acid was reported in several *Candida* spp. wild strains, as well as clinical isolates and mutant strains [90]. Poacic acid is a natural plant metabolite with promising antifungal potential, found in the lignocellulosic hydrolysates of grasses [91]. The sensibility of *Candida* spp. to poacic acid was shown to be regulated by the calcineurin pathway and did not correlate with in vitro β-glucan synthase activity [90]. Furthermore, another study demonstrated increased expression of *IFD6* gene in *Candida* spp. caused by neopodin, an anthraquinone from *Rumex crispus* [92]. *IFD6* encodes an aryl alcohol dehydrogenase considered to be an inhibitor of biofilm matrix production through quorum-sensing aryl and acyl alcohols [39]. This study opens up new perspectives in the search for the mechanisms whereby the plant metabolites alter the cell wall and biofilm composition.

Numerous cases of *Candida* resistance to antifungals regard the use of azoles responsible for cellular membrane destabilization. It has been demonstrated that diverse plant components might regulate ergosterol biosynthesis, including geraniol, a cyclic monoterpene alcohol present in diverse plant-derived essential oils that showed a satisfactory antifungal activity [93]. This terpene was able to inhibit the growth of *Candida* spp. at low MIC (Minimal Inhibitory Concentration) values. In that study, a geraniol concentration-dependent reduction of ergosterol content in treated fungal cells was observed, which was associated with the destruction of membrane integrity. In addition, an inhibitory effect on the proton pump H^+^-ATPase in the plasma membrane was detected after geraniol treatment and, as the conventional drugs—fluconazole and AMB—have no significant effects on H^+^-efflux, a synergistic action of geraniol with these antifungals was suggested. Recently, another plant-derived compound was reported to possess antifungal activity against non-albicans *Candida* species based on similar mechanisms—β-citronellol which was able to decrease H^+^-efflux rates in *C. glabrata* and *C. tropicalis* and to inhibit ergosterol biosynthesis [94]. Eucalyptal D, a formyl-phloroglucinol meroterpenoid, isolated from *Eucalyptus robusta,* was also tested for antifungal properties in wild strains as well as fluconazole-resistant clinical isolates of *C. albicans* [95]. This study showed that fluconazole-susceptible strains were vulnerable to eucalyptal D, while fluconazole-resistant strains also presented resistance against eucalyptal D; however, this compound might act synergistically with fluconazole in the inhibition of the growth of resistant strains. Although eucalyptal D was able to reduce fluconazole efflux in mutants strains, the up-regulation of the genes responsible for the expression of ABC transporters *CDR1* and *CDR2* was observed, suggesting that this substance may competitively inhibit fluconazole efflux due to its affinity to membrane transporters. On the other hand, essential oils of *Cinnamomum verum* and *Pelargonium graveolens* rich in terpenes and phenylpropanoids were able to act synergistically with fluconazole, resulting in membrane destabilization due to reduced ergosterol biosynthesis and disturbed fatty acid homeostasis in *C. albicans* cells [96]. Gupta and co-workers [89] also demonstrated the reduction of ergosterol synthesis in *C. glabrata* isolates by another terpene–cinnamaldehyde, due to a decreased expression of gene encoding enzymes involved in ergosterol synthesis in the cell membrane (*ERG2, ERG3, ERG4, ERG10, ERG11*) and gene encoding ABC transporters (*CDR1*).

Other plant-derived substances are also involved in membrane destabilization. Patel et al. [97] reported the inhibition of ergosterol biosynthesis in an inhibitor-concentration dependent manner, resulting in membrane disruption in cells treated with a flavonoid (5,6,8-trihydroxy-7,4′-dimethoxyflavone) derived from *Dodonaea viscosa* var. *angustifolia*. Furthermore, a *Syzygium cumini* methanolic extract exhibited antibiofilm activity against *C. albicans* without fungicidal activity against planktonic cells [98]. The major component of the extract fraction with the largest inhibitory properties against biofilm was quinic acid—a cyclic polyol. In that study, a synergistic effect on inhibition of biofilm formation by this fraction combined with undecenoic acid was also demonstrated. This effect correlated with a significant decrease in the polysaccharide and lipid content within the extracellular biofilm matrix. The mixed substances were responsible for the down-regulation of gene expression involved in ergosterol biosynthesis (*ERG11*) and in membrane transport (*CDR1*, *MDR1*, and *FLU1*), the expression of which is mostly up-regulated in azole-resistant *Candida* spp. [68]. Therefore, the inhibition of efflux pumps was proposed as one of the mechanisms to counteract drug resistance. In fact, it was demonstrated that the reduction of virulence of *Candida* spp. by quinic acid and undecenoic acid might be attributed not only to decreased ergosterol levels but also to the down-regulated expression of membrane transporters [98]. Berberine, a plant alkaloid identified in several plants from the *Berberis* genus presented antiproliferative effect in fluconazole-resistant *C. tropicalis* strains and improved its fungicidal activity [99]. Moreover, berberine was able to reinforce fluconazole efficacy in these strains via different mechanisms, including ergosterol decrease and efflux inhibition.

The effects of plant peptides on bacterial cell membrane perforation have been widely studied, and it can be expected that these peptides may have similar activity against *Candida* spp. Earlier reviews suggested that the number of plant-derived peptides with anticandidal activity was relatively modest compared with other peptide sources [100,101,102]. Antimicrobial peptides from plants include defensins, thionins, lipid transfer proteins, snakins, cyclotides, knottins and heveins [103]. They were demonstrated to act via the membrane permeabilization, inhibition of cell wall synthesis, binding to DNA or RNA, inhibition of DNA, RNA and protein synthesis, repression of protein folding and metabolic turnover, as well as induction of apoptosis [100]. Nevertheless, the mechanisms involved in their anticandidal action are still poorly understood. The last reports suggested membrane permeabilization as the main mechanism of plant peptide action. Among these peptides, the most studied have been defensins—short, cationic peptides with high cysteine content [104]. A defensin from *Heuchera sanguinea* (HsAFP1) and recombined radish defensins (RsAFP1 and RsAFP2) presented antifungal activity against planktonic *C. albicans* cells [105,106]. HsAFP1 showed a synergistic effect with caspofungin on inhibition of *C. albicans* biofilm, while together with AMB strongly inhibited the growth of the planktonic cells. It was also suggested that this defensin may also be involved in the inhibition of yeast-to-hypha transition, but the mechanism of this process remained undefined. Similar observations were described in another report, indicating a synergistic action of a smaller linear HsAFP1-derived peptide (HsLin06_18) with several echinocandins (caspofungin, micafungin and anidulafungin) against *C. albicans* [107]. These substances also significantly reduced in vitro biofilm formation on catheters by *C. glabrata* and caspofungin-resistant *C. albicans* strains. It was proposed that caspofungin facilitated HsLin06_18 internalization. Since plant defensins are in general nontoxic towards mammalian cells, these achievements suggest an attractive perspective of a novel candidiasis treatment. Furthermore, the plant-derived thionin from *Capsicum annuum* fruits (CaThi) exhibited toxic effects against *Candida* spp. causing membrane permeabilization and facilitating the entrance of fluconazole into the cell cytoplasm [108]. A protein from *Oryza sativa*, tetratricopeptide domain-containing thioredoxin (OsTDX)*,* also showed antifungal activity against *Candida* spp. via a destabilizing and disrupting effect on fungal membranes [109]. Mechanisms that are related to facilitation of entry or a limitation of the efflux of antifungal drugs have also been elucidated for plant-derived peptides [101]. A novel cysteine-rich peptide belonging to the heveins and isolated from *Pereskia bleo* (bleogen pB1) was reported to be an anticandidal agent and its heparin-binding property was indicated as essential for antifungal activity. However, this peptide was probably not membrane-lytic and at high-salt conditions, its anticandidal activity was lost, suggesting the importance of ionic interactions for bleogen pB1 antifungal action [110].

The above-described studies certainly allowthe affirmation that bioactive plant-derived substances may act as anti-*Candida* agents due to alteration of the cell wall and membrane reorganization, especially by decreasing polysaccharide and ergosterol synthesis, as well as by inhibiting drug efflux and altering membrane permeability and hydrophobicity.

### 4.2. Mechanisms of Antivirulence Activity of Plant Metabolites on Candida Species

In the last five years, different plant-derived compounds have been studied for the regulation of processes involved in fungal virulence-related mechanisms, such as adhesion, hypha formation or production of extracellular proteases and other hydrolases. As described previously, Muthamil and co-authors [98] demonstrated a synergistic effect of quinic acid with undecanoic acid against *Candida* spp. isolates. In this report, the authors presented additional mechanisms for antifungal activity, including the down-regulation of representatives of the *ALS* gene family (*ALS1*, *ALS3*) and decreased expression of *SAP1*, *SAP2*, and *SAP4* genes resulting in a remarkable reduction of extracellular aspartyl protease activity in treated *C. albicans.* Furthermore, in that study, the genes *HWP1, EGF1, CPH1, UME6, EAP1, RAS1, HST7*, and *CST20,* that are directly or indirectly involved in hyphal growth or filamentation, were found to be appreciably down-regulated by quinic acid/undecanoic acid combination. In contrast, increased expression of negative regulators of hyphal genes (*NRG1, TUP1)* was noticed. In a most recent report, the effect of oleic acid against *C. albicans* and *C. tropicalis* was also explored through transcriptomic and proteomic approaches [111]. This fatty acid presented good efficacy to inhibit biofilm formation and decrease the virulence of *Candida* spp. through the regulation of protein activities involved in glucose metabolism, ergosterol biosynthesis, lipase production, iron homeostasis and amino acid biosynthesis. In addition, a significant change in the expression of genes that are related to adhesion (*ALS1, ALS3, EAP1*), ergosterol biosynthesis (*ERG11*), SAP production (*SAP1, SAP2, SAP4*), filamentation (*HWP1, EFG1, CST 20*, *RAS1, UME6, HST7*) and efflux pump activation (*CDR1, MDR1*) was demonstrated. The antivirulence activity of teasaponin, a surfactant from tea seeds, was also recently proposed [112]. Teasaponin showed moderate antifungal activity against the wild type, efflux pump mutants and multidrug-resistant *C. albicans* strains. This metabolite inhibited hyphal formation via the Ras–cAMP–PKA signaling pathways, leading to the significant down-regulation of *RAS1*, *ALS3*, *HWP1*, *CDC35*, *EFG1*, *ECE1* genes.

Plant-derived phenolic compounds have recently attracted increasing interest in their antifungal activity. This large group includes numerous classes of compounds such as phenols, coumarins, lignans, flavonoids, anthocyanins, tannins, quinones and stilbenoids. In fact, magnolol and honokiol, two neolignan compounds isolated from *Magnolia officinalis* were able to inhibit adhesion, yeast-to-hypha transition and biofilm formation in *C. albicans* via Ras1-cAMP-Efg1 pathway [113]. The treatment of various *C. albicans* strains with these neolignans caused a significant decrease in the expression of genes activated by this pathway and the MAPK cascade pathway (*RAS1, EFG1, TEC1, CDC35, ECE1, HWP1, ALS3*) that play a key role in inducing adhesion of hyphal cells. Moreover, exogenous cAMP restored the hyphal formation in magnolol- and honokiol-treated strains. Similar conclusions were reached in another study, in which the alkaloid sanguinarine was shown to down-regulate some adhesion- and hypha-specific/essential genes related to cAMP, such as *ALS3*, *HWP1*, *ECE1*, *HGC1*, and *CYR1* in various *C. albicans* strains [114]. Antibiofilm and antihyphal activities were also induced by camphor and fenchyl alcohol, phenolic metabolites from *Cedrus libani* showing a down-regulation of gene expression related to hyphal and biofilm formation (*ECE1*, *ECE2*, *RBT1*, *EED1*) [115]. These authors also demonstrated that anthraquinone derivatives, such as alizarin (1,2-dihydroxyanthraquinone) and chrysazin (1,8-dihydroxyanthraquinone) could suppress *C. albicans* biofilm formation owing to a decreased expression of several hypha-specific and biofilm-related genes (*ALS3*, *ECE1*, *ECE2*, and *RBT1*) [116]. In other reports, it was demonstrated that shikonin, the major constituent of the red pigment extracts from the roots of *Lithospermum erythrorhizon* exerted an inhibitory effect on biofilm formation and mature biofilm destruction [117]. Several hypha- and adhesion-specific genes were differentially expressed in shikonin-treated biofilm, including the down-regulation of *ECE1*, *HWP1*, *EFG1*, *CPH1*, *RAS1*, *ALS1*, *ALS3*, *CSH1* and up-regulation of *TUP1*, *NRG1*, *BCR1*. In turn, phenylpropanoid cinnamaldehyde was able to decrease *HWP1* gene expression leading to the enhanced killing of *C. albicans* isolates and inhibition of biofilm formation [118]. Transcriptomic study on the action of 6-gingerol and 6-shoagol on *C. albicans* cells showed that these polyphenolic compounds repressed the expression of hypha/biofilm-related genes (*ECE1* and *HWP1),* thus explaining the observed cell growth primarily in planktonic form and a reduced adhesion [119]. The inhibition of *Candida* spp. biofilm formation by nepodin, a phenolic compound that diminished the hypha- and biofilm-related genes (*ECE1, HGT10, HWP1, UME6*) was also confirmed [92]. Furthermore, a flavonoid belonging to the chalcone group―loureirin A, the major active component of *Draconis sanguis*, down-regulated the expression of adhesion-related genes and hypha/biofilm-related genes (*HWP1*, *IFF4*, *EAP1*, *ECE1*, *RAS1*, *GPR1*, *CYR1*, *EFG1*, *CEK1*, *CPH1*, *UME6*, *PGA10*) [120].

Other secondary metabolites from the terpene group also have been studied for their antifungal activity. Coumarin, a natural plant terpene, strongly affected the capacity of *C. albicans* to form biofilm and significantly impaired the preformed mature biofilm [121]. The expression of adhesion- and hypha-related genes, including *HWP1, HYR1, ECE1*, and *ALS3*, was remarkably down-regulated by coumarin. Moreover, the addition of exogenous cAMP partly returned the coumarin-induced inhibition of hyphal development. Another compound of this group, zerumbone, a monocyclic sesquiterpene extracted from *Zingiber zerumbet* significantly down-regulated the expression of biofilm-related and hypha-specific genes (*HWP1* and *ALS3),* causing the inhibition of biofilm formation and eradication of preformed biofilm of *C. albicans* isolates [122].

In the last decade, new mechanisms for the regulation of *Candida* pathogenesis have been proposed. The survival of *Candida* cells depends on their ability to adapt to adverse environmental conditions. It has been shown that quorum-sensing molecules may play an important role in the regulation of the fungal adaptation to environmental changes. For instance, regulation of apoptotic processes in *C. albicans* cells by farnesol was demonstrated, suggesting that this quorum-sensing molecule is involved in the regulation of the fungal cell life cycle with important implications for adaptation and survival [123]. For this reason, new ideas have emerged to elucidate the mechanisms responsible for the effectiveness of plant metabolites for the control of *Candida* pathogenesis. In fact, numerous studies indicated the mediation of plant metabolites in pro-apoptotic processes, such as increased ROS release, disruption of mitochondrial membrane potential, phosphatidylserine externalization, cytochrome C release, metacaspase activation, DNA damage and nuclear fragmentation.

Cinnamaldehyde―a phenylpropanoid mentioned above for its anticandidal activity, also induces pro-apoptotic processes. It was demonstrated that wild and isolated strains of *C. albicans* [124] and *C. glabrata* [89] treated with cinnamaldehyde presented increased ROS production and increased release of cytochrome C from mitochondria. Further effects observed in that study included an enhanced calcium concentration in the cytoplasm and mitochondria, activation of metacaspase, DNA damage, and chromatin condensation when membrane phosphatidylserine was exposed [124]. In another study, the effect of cinnamaldehyde, free or encapsulated in multimicellar liposomes, on biofilm formation by *C. albicans* was examined [118]. In both cases, pro-apoptotic effects were observed, leading to the inhibition of biofilm formation. However, these authors stated that the unencapsulated metabolite, which showed lower efficacy, was involved in early and late apoptosis, while the encapsulated compound induced early apoptosis and necrosis. Another phenylpropanoid, associated with the induction of apoptosis in *C. albicans* cells, was hibicuslide C, extracted from *Abutilon theophrasti* [125]. This compound exerted antifungal activity against *C. albicans* through the induction of early apoptosis followed by secondary necrosis. Hibisculide C induced an increase in ROS production with the accumulation of hydroxyl radicals, which was accompanied by mitochondrial dysfunction, metacaspase activation, and an increase in intracellular Ca^2+^ level. Furthermore, flavonoids have been shown to be associated with apoptosis induction in *Candida* spp. Wang and co-authors [126] reported that baicalin from *Scutellaria* might suppress the development of *C. albicans* biofilms most likely due to inducing cell death via apoptosis. This flavonoid inhibited *C. albicans* biofilm through extensive chromatin condensation, ROS accumulation, phosphatidylserine externalization, nuclear fragmentation, metacaspase activation and cytochrome C release. In addition, the expression of genes that are closely related to the Ras-cAMP-PKA pathway which has been considered responsible for apoptosis regulation in *C. albicans* was up-regulated (*RAS1*, *TPK1*) or down-regulated (*CAP1*, *PDE2*). Silibinin, the most active constituent of *Silybum marianum* extract, is also a flavonoid with anticandidal activity exerted through the apoptotic mechanism [127].

For berberine, a herbal-derived alkaloid, antifungal activity tests against fluconazole-resistant *C. tropicalis* isolates showed that this compound was involved not only in the modification of ergosterol synthesis and efflux pump efficiency, but its antifungal properties were also related to apoptosis of fungal cells, as documented by the berberine-induced ROS production [99]. Moreover, this effect was more remarkable when cells were treated with berberine and fluconazole, indicating that the alkaloid might be helpful in the treatment of fluconazole-resistant *C. tropicalis* strains. Another plant-derived metabolite, lycopene, a carotenoid found in various fruits and vegetables was able to induce apoptotic processes, employing antifungal activity during the early and late stages of apoptosis in *C. albicans* cells [128].

The roles of proteins, peptides, and other primary metabolites in *Candida* pathogenicity attenuation has also been reported. The role of the Kunitz type trypsin inhibitor from *Cassia leiandra* seeds (*Cl*TI) in the growth inhibition of *Candida* spp. was presented [129]. *Cl*TI anticandidal activity was related not only to the alteration of membrane permeabilization and H^+^-pump activity, but also to the induction of oxidative stress and DNA damage, suggesting the involvement of apoptotic and necrotic mechanisms. A similar mechanism was proposed for the antifungal activity of the thionin-like peptide (*Ca*Thi) from *Capsicum annuum* fruits [108]. *Ca*Thi caused plasma membrane permeabilization in all yeasts tested but induced oxidative stresses only in *C. tropicalis*. In addition, *Ca*Thi acted synergistically with fluconazole, inhibiting all tested *Candida* spp., achieving even complete inhibition of *C. parapsilosis* growth.

Research on the anticandidal activity of lectins is also currently gaining an interest. A lectin found in *Punica granatum* juice—PgTeL—has been found to be an antifungal agent against *C. albicans* and *C. krusei* [130]. *Candida* treatment with PgTeL resulted in a decrease of intracellular ATP and induced lipid peroxidation, suggesting a mechanism that involves oxidative stress, energetic collapse, damage of the cell wall, and finally rupture of yeast cells.

In addition to the plant-derived products previously described, a number of further plant extracts with antifungal activity have been reported (see Appendix A).

The studies outlined herein provided novel information on the antifungal activity of plants, with emphasis on the mechanisms underlying the effects of plant metabolites on the inhibition of the growth of *Candida* spp. and impairment of the candidal virulence. The discovery of these mechanisms has opened up new perspectives for the treatment of candidiasis, especially in the problem of eradication of drug-resistant strains. The mechanisms responsible for the resistance to antifungal agents are continuously studied and it was established that drug resistance in *Candida* spp. is a complex multifactorial phenomenon, involving molecular mechanisms often different for planktonic cells and for biofilm. Therefore, the use of the traditional antifungal drug in combination with plant metabolites can offer a new approach to candidiasis prevention and treatment. Such an approach might expand the spectrum of antifungal activity and increase safety and tolerance due to the lower doses of applied drugs. It should also be noted that in the majority of the reports describing the cytotoxicity of the plant metabolites towards fungal cells, no adverse effects on mammalian cell viability have been shown. However, almost all related studies were performed in vitro; only in some cases, experiments have been carried out on animal models, clearly meaning that these effects require further confirmation.

In conclusion, plants may be considered as a valuable source of bioactive substances that can help in the fight with drug-resistant *Candida* strains and their exploration is highly recommended to offer new effective antifungal therapies.

## Figures and Tables

**Figure 1 ijms-21-06131-f001:**
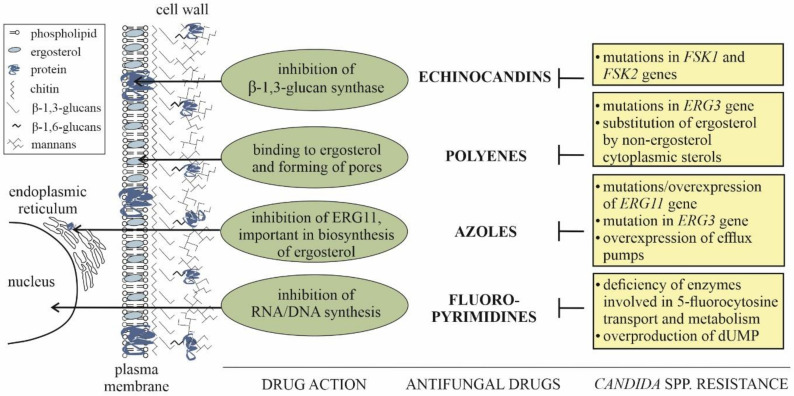
Currently used groups of antifungal drugs with an indication of their cellular targets and the mechanisms of resistance developed by *Candida* spp.

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
