# Peer review of "Plant-Derived Substances in the Fight Against Infections Caused by *Candida* Species"

_ijms, 2020, doi:10.3390/ijms21176131_

Round 1
Reviewer 1 Report
The review by Guevara-Lora et al. deals with the interesting issue of plant-derived substances active against Candida species.
Principal issues
Lines 199-201: “Especially in the last five years, new achievements in this field demonstrate a great interest in the anti-Candida activity of plant-derived crude extracts or isolated phytochemicals (Supplementary Table 1).” It is not clear why reference to Supplementary Table 1 is reported at the end of this sentence. Since plants extracts reported in the supplementary table are not cited in the text and the related references – 131-169 – are the last of the reference section, it is suggested to delete the reference to supplementary table in this point. The sentence should be: “Especially in the last five years, new achievements in this field demonstrate a great interest in the anti-Candida activity of plant-derived crude extracts or isolated phytochemicals.” It is also suggested to add a sentence later in the text, i.e. on line 468, before the final considerations, indicating that, in addition to the plant-derived products previously described, a number of further plant extracts with antifungal activity have been reported (see Supplementary Table 1).
Lines 208-209: “Although a great number of reports have demonstrated the effect of plant extracts during candidiasis treatment, …” as correctly stated on lines 480-481, almost all reported studies on antifungal activity of plant extracts were performed in vitro, so the sentence could be rewritten as follows “Although a great number of reports have demonstrated the antifungal effect of plant extracts, …”
Lines 231-234 and lines 288-298: it is suggested to combine the sentences that refer to the same paper.
Minor issues
Abstract
Line 33: “…merging Candida strains…” should it be “…emerging Candida strains…”?
Introduction
Line 47: “…fungal pathogenic in humans…” should it be “…fungal pathogens in humans…”?
Line 52: “…yeast-like fungi cause of a wide…” should it be “…yeast-like fungi cause a wide…”?
Line 99: “Another virulence trait is exerted by secreted extracellular hydrolases, e.g. phospholipases that degrade phospholipids of host cell membranes and facilitate their penetration by fungal cells.” it should be better to write “Another virulence trait is related to secreted extracellular hydrolases, e.g. phospholipases that degrade phospholipids of host cell membranes and facilitate tissue penetration by fungal cells.”
Line 237: Paeonia lactiflora
Line 267: β-citronellol
Line 287: Dodonaea viscosa
In Reference 80 the year of publication is not reported.
Please check genus and species names in Reference section, some of them are not in italics (e.g. ref 98, 108, 165, 166…)
Author Response
We would like to thank the reviewer for the comments made, which allowed us to prepare a better version of the manuscript.
The awkward wording in the stated items were corrected in accordance with the reviewer's recommendations.
1 / the references concerning data included in table 1 were moved from the main list of references in the manuscript to the file in supplementary materials containing Table 1. The references cited in table 1 received new numbering.
2/ in all references listed, the genus and species names were corrected (italic letters were used where it was necessary)
2 / line 199-201: we removed the information about supplementary materials and moved it, as recommended by the reviewer, to line 468. The change was marked in yellow color.
3 / the text from lines 231-234 was removed and merged with the information in lines 288-298; the introduced change was marked in yellow color
4 / any minor changes in the text and typos were corrected as suggested by the reviewer, with the change marked in yellow color
Reviewer 2 Report
An interesting and valuable review of a scientific area that has been somewhat neglected: The patogenic activities of various Candida species. The rapid development of resistance against our most important antifungal drugs used to treat Candida infections is a huge problem, and the search for novel drug candidates in Nature is part of the solution. The manuscript describes the various ways to interfere with the Candida cells on a molecular level and how this can be modulated by novel secondary metabolites, is a very important starting point for novel treatments, and after a slight linguistic polish this manuscript is acceptable for publication.
Author Response
We would like to thank the reviewer for the comments made, which allowed us to prepare a better version of the manuscript.
1/ any minor changes in the text and typos were corrected as suggested by the reviewer, with the change marked in yellow color;
2 / the changes made during language correction, as suggested by the reviewer, are also marked.